# Analysis of Calving Ease and Stillbirth and Their Impact on the Length of Functional Productive Life in Slovak Holstein Cattle

**DOI:** 10.3390/ani13091496

**Published:** 2023-04-27

**Authors:** Eva Strapáková, Juraj Candrák, Peter Strapák

**Affiliations:** 1Institute of Nutrition and Genomics, Faculty of Agrobiology and Food Resources, Slovak University of Agriculture in Nitra, Trieda Andreja Hlinku 2, 949 76 Nitra, Slovakia; 2Institute of Animal Husbandry, Faculty of Agrobiology and Food Resources, Slovak University of Agriculture in Nitra, Trieda Andreja Hlinku 2, 949 76 Nitra, Slovakia

**Keywords:** Holstein cows, longevity, calving ease, stillbirth, survival analysis

## Abstract

**Simple Summary:**

One of the important goals of cattle breeding is increasing the longevity of cows, which is one of the most important economic traits. In our work, we focused on investigating the effects of factors that could be the cause of cow culling before they reach their maximum potential. Calving difficulty and the proportion of stillborn calves reduce the length of the productive life because cows with problematic calving are often culled from the herd. For the farmer, these traits represent additional costs for veterinary procedures and related treatment after calving. We found that factors such as milk production level, parity, age at first calving, herd size, calf sex, herd–year–season, calving ease, and stillbirth had a significant effect on the longevity of cows. The research confirmed that the incidence of difficult births and the proportion of stillborn calves was higher in primiparous cows than in multiparous cows. Our analysis showed that female calves were born more easily than male calves. By selecting animals aimed at reducing the number of difficult calving and the number of stillborn calves, a higher functional, productive life in cows could be achieved.

**Abstract:**

The aim of the study was to determine the frequency of births according to the categories of calving difficulty and stillbirths and to evaluate the effect of these factors on the longevity of cows. Longevity is one of the traits that affect the overall profit in the dairy industry. A Weibull proportional hazard model was used to evaluate the influence of functional traits such as calving ease and stillbirth. Longevity was expressed as the length of a functional, productive life from the first calving to death or censoring, which was corrected for milk yield. The database included 918,568 calvings, where calving without assistance represented 83.34%, calving with the assistance of one person or the use of a slight mechanical pull represented 14.47%, difficult calving with the assistance of several people, the use of mechanical traction or the intervention of a veterinarian represented 2.16%, and cesarean section represented 0.03%. The mortality of calves, stillborn or dead within 48 h of birth, represented 1.07% and 6.59%, respectively. The frequency of alive female calves was higher (46.84%) than male calves (45.50%). Cows with higher lactations had almost half as many stillborn calves as heifers. The most stillborn calves were found in difficult births (59.48%). In easy calving, this proportion was 2.48%. Using survival analysis, we estimated the significant influence of the factors such as parity, milk production, herd size, age at first calving, herd × year × season, sex of calf, calving ease, and stillbirth on the length of the functional, productive life of cows. The risk of early culling of the cows with moderately difficult calving was 1.259 times higher than in the cows with easy calving. Difficult calving and cesarean section shorten the productive life, and the risk of culling reached 1.711 and 1.894, respectively. Cows that gave birth to a dead calf achieved a 2.939 times higher risk of culling compared to cows that gave birth to a live calf. In this study, a higher risk of early culling was found in cows that gave birth to a male calf. Evaluation of the calving ease and stillbirth can be used as indirect indicators at an earlier age of the animal in the selection process for long-lived animals with good productive and reproductive performance.

## 1. Introduction

In the past, selection indices were created for dairy cows to increase milk production and milk components without regard to other functional traits. This selection method caused unfavorable genetic responses for traits such as longevity, reproductive performance, fertility, and the incidence of various diseases [1]. Therefore, it was necessary to supplement the selection criteria with additional important economic traits that indirectly affect animal production costs and mitigate the negative impact of high milk yield on the above traits [2]. Several authors point to a negative relationship between milk production and longevity, health, and calving ability [2,3,4]. Longevity is the most important functional trait in dairy cattle. Longevity directly affects reducing the costs for the replacements of cows in the herd and contributes to an increase in the average herd yield through an increase in the proportion of cows in the higher parities [5]. Longevity is expressed through several indicators, such as the time period from birth to death, productive life from the first calving to death or culling (voluntary culling due to low milk production) [6], stayability up to a certain period of time [7], functional productive life from the first calving to culling, or death caused by involuntary culling due to disease or infertility [8,9]. Because the heritability of longevity is low (h2 = 0.01–0.3) [10], its improvement is mainly influenced by environmental factors such as herd management, living conditions, housing, nutrition, climatic conditions, and others [11,12,13]. Evaluation of direct longevity is complicated because the recording of phenotypic data is performed late in the life of the animal. Furthermore, the data have a skewed distribution. Survival analysis using a Weibull proportional hazards model can offer a better fit to survival data due to its ability to correctly account for the records of the still-alive cows (censored data) at the time of analysis [8]. Longevity can also be estimated indirectly, at an earlier age, on the basis of correlated traits, such as udder, feet and legs traits [14,15], body condition score [16], calving ease, and the number of stillborn calves [10,17,18,19].

Calving difficulty is one of the risk factors in terms of culling [20] because it often results in additional breeding costs and creates a greater need for the breeder work and increases veterinary costs, risk of disease, and calf loss or reduces milk yield [21,22,23]. Calving is affected by the size of the calf and parity and additional factors, such as gestation length, age at first calving, and calving interval [24]. Several authors confirmed the relationship between difficult calving and calf size and cow parity. In primiparous cows, larger calves showed a higher mortality rate than in cows of higher parity. This is primarily related to the size of the birth canals, which are less restrictive in older cows. It is easier to give birth to larger, better-developed calves, which would be more difficult in primiparous cows [17,25,26].

Difficult calvings reduce the length of a productive life by 10% [21], and the risk of cow culling in difficult births increases in the first 30 days after calving [19,27]. The course of calving is very often an important predisposing factor for later appearing fertility disorders [28].

Based on this knowledge, the work aimed to determine the frequency of calving according to the categories of calving difficulty and stillbirth and to evaluate the effect of calving ease and stillbirth on the longevity of Slovak Holstein cows.

## 2. Material and Methods

### 2.1. Longevity

Longevity, calving ease, and stillbirth records of the Slovak Holstein cows calved from 1996 to 2022 were provided by the Slovak Breeding Services, s.e.(Bratislava, Slovakia)。

The database contained 716,576 cows of 6 parities with a known date of 1st calving. Another criterion for the inclusion of cows in the evaluation was the age at calving, from 600 to 1200 days. The cows belonged to 1661 herds.

Uncensored records consisted of cows that were marked as culled (death) by the time of analysis. Cows alive at the time of analysis, cows with milk yield less than 1700 kg (unfinished lactation), cows that reached more than 5th lactation, cows removed from the Slovak National Milk Recording System, or cows sold for dairy purposes and herds with less than 10 calvings per year were marked as right-censored records. These data were censored due to incompleteness or the small number of observations in individual classes at the time of analysis. Longevity was expressed as the length of the functional productive life because it better describes real longevity, which is not affected by voluntary culling due to low milk production. The length of productive life was measured in days and represents the time period from the first calving to death or censoring.

SAS 9.2, Enterprise Guide 4.2 was used to prepare the database [29].

### 2.2. Calving Ease and Stillbirth

From the total number of cows in the longevity database, only 451,473 cows (63%) had available data on calving ease and stillbirths. There were 918,568 calvings recorded. Based on the methodology for evaluating the progress of births in Slovakia, twin calves were omitted from the data. Calving ease was categorized into 5 groups: 0, missing data; 1, easy calving (without assistance); 2, moderately difficult calving (the assistance of one person or the use of a slight mechanical pull); 3, difficult calving (the assistance of several people, the use of mechanical traction or the intervention of a veterinarian); 4, operation (cesarean section, fetotomy). Three categories of the stillbirth of a calf were defined: 0, missing data; 1, alive; 2, stillbirth (stillborn or died within 48 h). The Slovak recording system does not require recording the sex of the stillborn calf. This represents a crucial point regarding problems in the correct assessment of the calf sex effect. Based on the methodology for estimating breeding values for calving ease in Slovakia, 5 calf sex classes were created: 0, missing data; 1, male; 2, female; 16, sex unspecified–stillborn; 61, sex unspecified–died within 48 h.

Missing data (37%) were included in the analysis to reduce bias.

### 2.3. Model

The influence of selected traits on the longevity of cows was evaluated using Survival analysis [30]. The following Weibull proportional hazard model was used. The impacts of calving ease and stillbirth on longevity were estimated separately, and two analyses were carried out.

Model:λ(t)_1,2_ = λ0 (t) exp(P + M + HS + age + HYS + T_1,2_ + sex)
where:

t is time in days from the first calving to culling or censoring data

λ(t) is the hazard function for a given cow at time t

λ0(t) is the Weibull baseline hazard function describing the aging process that assumes a Weibull distribution with two parameters λ and ρ.

P is the fixed time-dependent effect of parity (5 classes with changes at each calving date).

M is the fixed time-dependent effect of the milk production class, expressed as a standard deviation (SD) from within-herd-year average (5 classes: 1, M > +2 SD from herd-year average; 2, m ≥ 1 SD and m ≤ +2 SD; 3, M > −1 SD and M < +1 SD; 4, M ≤ −1 SD and M ≥ −2 SD; 5, M < −2 SD).

HS is the fixed time-dependent effect of the herd size variation was expressed as a change in the current cows’ ratio in the herd compared to the previous year (6 classes: 1, HS < −75%; 2, HS from −75% to <−25%; 3, HS from −25% to <0%; 4, 0%; 5, HS from >0% to <+25%; 6, HS ≥ +75%).

Age is a fixed time-independent effect of the age at first calving (5 classes: 1, age from 600 to 720 days; 2, age 721 to 840 days; 3, age from 841 to 960 days; 4, age from 961 to 1080 days; 5, age from 1081 to 1200 days).

HYS is the random time-dependent effect of the herd × year × season interaction, following a normal distribution. The effect covers the years 1996–2022 and the two seasons, April–September and October–March.

T_1_ is a fixed time-dependent effect of calving ease (5 classes: 0, missing data; 1, calving without assistance; 2, calving with assistance; 3, difficult calving; 4, cesarean section).

T_2_ is a fixed time-dependent effect of stillbirth (3 classes: 0, missing data; 1, alive calf; 2, stillborn, including dead calves within 48 h of birth).

Sex is a fixed time-dependent effect of the calf sex (5 classes: 0, missing data; 1, male; 2, female; 16, stillborn; 61, died within 48 h of birth; 16 and 61 are classes without sex determination).

The influence of individual factors on longevity was tested by the likelihood ratio tests, which compared the full model with the restricted model by excluding one effect under testing at a time. The coefficient of determination R^2^ of Maddala [31] was used as a measure of the proportion of explained variation by the model:R^2^_M_ = 1 − (LR/LU)^2/n^
where n is the total sample size and LR a LU is the restricted and unrestricted (full model) maximum likelihoods.

The influence of individual factors on longevity was expressed as the relative risk of early culling (RRC). The classes with the highest number of culled cows represented the reference level, and the RRC was set to 1.

## 3. Results and Discussion

The frequencies of calving ease according to the sex of the calf born are shown in Table 1. From the total number of calvings (918,568), 83.34% represented easy calvings, and 14.47% represented calvings with the assistance of one person. Difficult calving and cesarean sections reached a share of 2.16% and 0.03%, respectively.

In unassisted calving, 88.65% of female and 86.69% of male calves were born. Difficult calving occurred at a higher rate in the birth of a male than in the birth of a female calves. Spontaneous calving without assistance in the case of stillborn or dead calves within 48 h (unknown calf sex), represented 50.57% and 27.74%, respectively. The share of calving assisted by one person was higher by 12.84% for the group of dead calves within 48 h compared to the group of stillborn calves (Table 1).

Similar conclusions were reached by Barrier and Haskell [32], who reported 81.7% and 84.1% of easy calving in two smaller herds, and Sewalem et al. [18], who found 60.33% easy births in a population of Holstein cows in Canada. Morek-Kopeć et al. [17] stated 48.9% fewer unassisted calving in the Polish Holstein population and also a higher proportion of difficult births compared to our results. Similar results were also published in the Basque Holstein population [21]. Ryba [22] found 71.3% of easy calving and 2.62% and 0.04% of difficult births and cesarean sections in Slovak Holstein cows. More than 3.5% of difficult births and C-sections were reported in the Polish dairy cattle population [33]. One of the causes of difficult births can be the calf weight (males reached, on average, a higher weight than females) and the gestation length [34]. Morek-Kopeć et al. [17] found a 2.18 times higher risk of culling in the difficult birth of males and a 1.26 times higher risk of culling in the difficult birth of females in primiparous cows compared to unassisted calving.

Parity is one of the factors that affect calving ease (Table 2). The older cows in the 2nd parity achieved a 9.68% higher frequency of unassisted calving compared to primiparous cows. The assistance of one person was necessary for 19.40% of calving in primiparous cows and 11.37% of the cow calving in the 2nd parity. A decreasing trend between parities was found in difficult calving and cesarean sections. The percentage of cows in the individual categories of calving ease at the 2nd and higher parities reached approximately the same level (Table 2).

According to available scientific literature sources, older cows generally have fewer calving problems compared to heifer calving [24,35,36,37]. In recent years, difficult calvings have also occurred due to the selection pressure to include young heifers that have not reached their mature size at the first calving [23]. Heins et al. [35] found a decrease in calving difficulty by 8% in the 2nd parity compared to the 1st parity. Sawa et al. [38] reported an increase in unassisted births by 14.25% and, at the same time, a decrease in both easy and difficult births in multiparous cows in comparison with the primiparous cows. Pogorzelska and Nogalski [33] confirmed the same tendency of an increase in unassisted births in the 2nd parity by 13.28% and a linear decline in assisted births and difficult births by 10.88% and 3%, respectively, compared to heifers. Many authors [24,36,39] confirm the influence of age at first calving on the incidence of difficult calvings. It is probably related to the insufficient development of the reproductive tract and the ratio between calf size and feto–pelvic incompatibility [24].

Table 3 displays the mortality of calves by sex. The frequency of born and alive female calves was 1.34% higher than that of male calves. Stillborn calves (1.07%) and those that died within 48 h (6.59%) were without sex identification. The cause of stillbirths is more difficult to explain because stillborn calves were born in both easy and difficult calving.

Steinbock et al. [39] stated that 40–60% of all stillborn calves were born during normal calvings, which is more compared to our results. Many authors found significantly higher stillbirth rates in male than female calves [22,35,39]. Morek-Kopeć et al. [17] stated 7.65% and 3.67% of stillborn male and female calves in Polish Holstein-Friesian cows. The authors calculated overall stillborn calves on the level of 5.79%.

The evaluation of calf mortality according to parity is shown in Table 4. The ratio of alive to dead calves was found to be 90.01% to 9.99% for heifers and for cows in the 2nd parity 94.14% to 5.86%. This trend in the 3rd and subsequent parities had a slowly decreasing tendency.

Our results were confirmed by the studies of other authors, who report a higher proportion of stillbirths in heifers compared to older cows [17,34,35]. In heifers, there was a tendency to reduce the age at first calving, which had a negative effect on the proportion of stillborn calves. Steinbock et al. [39] stated a significant difference in the stillbirth rate in male calves between the calving of younger and older heifers. At the age of 26 months at the first calving, stillbirth reached 10 to 14%. At a higher age at the first calving, the stillbirth decreased to 8%.

Figure 1 displays the evaluation of stillbirth according to the categories of calving ease. The most stillborn calves were found in difficult calvings (59.48%), followed by C-section (54.61%) and moderately difficult calving (27.58%). The smallest proportion was found in easy calvings (2.48%).

Compared to our results, Sewalem et al. [18] and Morek-Kopeć et al. [17] stated a higher percentage (5.2% and 4.0%) of stillborn calves in calving without assistance category. Because stillbirth heritability is very low [40], improving this trait could be achieved through better herd management practices before, during, and after parturition [32]. It is also important to comply with the conditions set to achieve the breeding maturity of the heifers at the first insemination to avoid an increase in the number of stillborn calves and calving difficulties [39].

### Survival Analysis

Survival analysis was used to evaluate the influence of selected factors on longevity. The shape parameter ρ = 1.53912 and the scale parameter ρ log λ = −8.71238 were estimated in the first analysis. Similar parameters were reached in the second analysis. A value of ρ > 1 indicates that the culling risk increases with time and vice versa. Similar parameters were stated by Sewalem et al. [41] and de Maturana et al. [21], and higher values of 2.45 and 2.15 were detected by Caetano et al. [42] and Morek-Kopeć et al. [17].

In this study, 12.51% of right-censored records were used. The ratio between right-censored and uncensored data was 861 and 917 days from the first calving to censoring or dead cows (Table 5).

Many authors stated much higher censoring in Survival analysis, which may lead to a less reliable estimation of longevity breeding values [9,17,41,42,43] due to the use of more incomplete data.

All factors reached a highly significant effect (*p* < 0.01) on the length of the functional productive life (measured by −2logL Change) in both analyses (Table 6). The milk yield achieved the highest change in −2logL (633,508.7 and 637,354.8, respectively). The lowest −2logL Change (943.5) was found in age at first calving in the first analysis. The change in −2logL estimated for stillbirth was approximately two times higher than in calving ease. A significant effect of the sex of the born calf on the length of the productive life was also estimated (Table 6).

The influence of calving ease on the length of functional productive life was significant (Table 6). The risk of early culling in the cows with moderately difficult calving was 1.259 times higher than in the ones with easy calving (Figure 2A). Difficult calving and cesarean section resulted in a decrease in the length of functional productive life. The risk of early culling reached 1.711- and 1.894-times higher values for difficult calving and cesarean section, respectively, compared to easy calving. Except for missing data, the lowest risk of early culling (2.244) was achieved by births of female calves. In comparison with the male calves born, this represented a 1.08 times lower risk of early culling (Figure 2B). The highest risk of early culling (2.541) was found in cows whose calves died within 48 h after birth.

Figure 3A,B show the effects of stillbirths and the sex of calves on the length of a functional productive life. Cows that gave birth to a dead calf had a 2.939 times higher risk of early culling compared to cows that gave birth to a live calf (Figure 3A). In this analysis, it was confirmed that cows that gave male birth reached a shorter functional productive life compared to cows that gave female birth.

Our results were also confirmed by the findings of other authors. De Maturana et al. [21] stated that a greater calving ease score resulted in a greater culling risk. Cows with difficult calving and cesarean sections had a culling risk 1.18 times higher than cows with unassisted calving. Using the Weibull model, Sewalem et al. [18] found that cows with difficult calvings and cesarean section were at a higher risk of culling by 30 and 90% compared to calvings without assistance. A similar tendency is reported by Rostellato et al. [44]. Probo et al. [19] calculated the risk of culling cows that gave birth to bulls at 1.39 times higher than those that gave birth to heifers. The authors stated that multiparous cows with assisted births were 1.37 times more likely to be culled than the cows with unassisted births. In primiparous cows, this risk was even higher. In a population of Holstein-Friesian cows in Poland, the risk of culling cows caused by the stillbirth calf was highest for the first parity calving of bulls [17].

## 4. Conclusions

Difficult calving and calf mortality are one of the main risk factors that are the reason for culling cows. Especially in heifers, difficult births and stillborn calves occur more than in multiparous cows. The results show that difficult births and stillbirths were higher in the delivery of male calves category. Based on the results of this research, difficult calving and a high proportion of stillbirth calves result in a decrease in the length of a functional productive life. Cows that gave birth to male calves appeared to be at a higher risk of early culling. The study shows that calving ease and stillbirth could be suitable traits for predicting longevity in the earlier age of cows.

## Figures and Tables

**Figure 1 animals-13-01496-f001:**
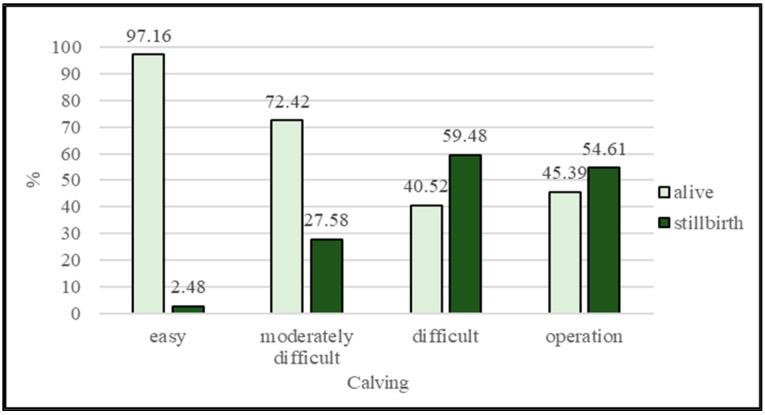
The ratio of alive and stillbirth calves by calving ease classes.

**Figure 2 animals-13-01496-f002:**
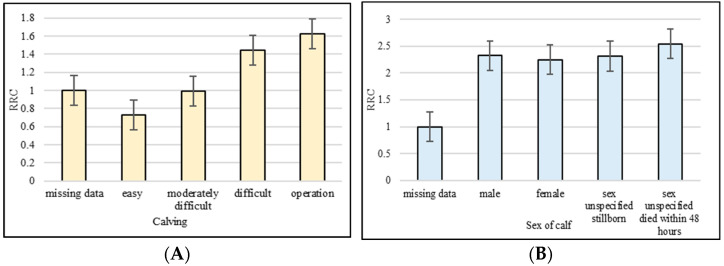
Relative risk of culling (RRC) for calving ease classes (**A**) and sex of calf (**B**). (Missing data on calving ease and sex of calf were set to 1).

**Figure 3 animals-13-01496-f003:**
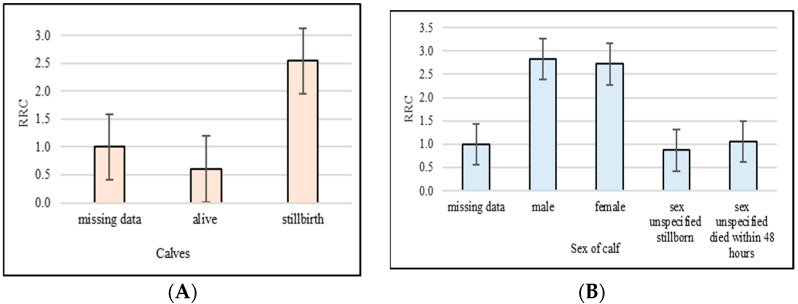
Relative risk of culling (RRC) for stillbirth (**A**) and sex of calf (**B**). (Missing data on stillbirth and sex of calf were set to 1).

**Table 1 animals-13-01496-t001:** Frequency of calving ease by sex of the calf.

Calving Ease	Sex of Calf		Unspecified Sex of Calf		Total
	Male	Female	Stillborn	Died within 48 h of Birth	
Easy calving	362,291 86.69%	381,441 88.65%	4983 50.57%	16,795 27.74%	765,510 83.34%
Moderately difficult	50,719 12.13%	45,561 10.59%	4044 41.04%	32,618 53.88%	132,942 14.47%
Difficult	4830 1.16%	3211 0.75%	823 8.35%	10,981 18.14%	19,845 2.16%
Operation	73 0.02%	50 0.01%	3 0.03%	145 0.24%	271 0.03%
Total	417,913 100%	430,263 100%	9853 100%	60,539 100%	918,568 100%

**Table 2 animals-13-01496-t002:** Frequency of calving ease classes by parity of dam.

Calving Ease		Parity of Dam					Total
	1	2	3	4	5	6	
Easy calving	268,914 77.47%	218,5678 7.15%	141,160 87.09%	80,434 86.75%	39,208 85.87%	17,227 85.34%	765,510 83.34%
Moderately difficult	67,328 19.40%	28,527 11.37%	18,443 11.38%	10,595 11.43%	5530 12.11%	2519 12.48%	132,942 14.47%
Difficult	10,736 3.09%	3645 1.45%	2451 1.51%	1663 1.79%	911 2.00%	439 2.17%	19,845 2.16%
Operation	149 0.04%	54 0.02%	32 0.02%	25 0.03%	10 0.02%	1 0.005%	271 0.03%

**Table 3 animals-13-01496-t003:** Frequency of calf mortality by sex of alive and stillbirth calf.

Trait	Alive		Stillbirth	Died within 48 h of Birth
Mortality of calf	**Male**	**Female**	**Unspecified Sex**	**Unspecified Sex**
	417,913 45.50%	430,263 46.84%	9853 1.07%	60,539 6.59%

**Table 4 animals-13-01496-t004:** Frequency of calf mortality by parity of dam.

Calf Mortality	Parity					
	1	2	3	4	5	6
Alive	312,442 90.01%	236,094 94.14%	151,957 93.75%	86,581 93.38%	42,392 92.84%	18,710 92.69%
Stillbirth or died after calving	34,685 9.99%	14,699 5.86%	10,129 6.25%	6136 6.62%	3267 7.16%	1476 7.31%

**Table 5 animals-13-01496-t005:** Descriptive statistics for length of functional productive life (*N* = 716,576).

	*N*	Min Time (Days)	Max Time (Days)	Average Time (Days)
Right censored records	89,674 12.51%	1	4891	861
Uncensored records	626,90 287.49%	1	6276	917

*N*—number of observation.

**Table 6 animals-13-01496-t006:** Comparison of the full model with the reduced models excluding one effect at a time using the Likelihood ratio test.

Effect	Analyse 1				Analyse 2			
	Δdf	−2logL Change	*p*	R^2^ of Maddala	Δdf	−2logL Change	*p*	R^2^ of Maddala
Milk yield	5	633,508.7	<0.01	0.1190	5	637,354.8	<0.01	0.1484
Parity	4	12,118.4	<0.01	0.7508	4	13,813.1	<0.01	0.6433
Change in herd size	5	12,670.8	<0.01	0.7604	5	12,752.7	<0.01	0.6438
Age at first calving	4	943.5	<0.01	0.7651	4	987.9	<0.01	0.6496
Calving ease	4	6688.7	<0.01	0.7652				
Stillbirth					2	12,255.7	<0.01	0.6441
Sex	4	22,608.6	<0.01	0.7653	4	35,279.4	<0.01	0.6324
Herd × year × season	*	99,839.2	<0.01	RANDOM	*	100,005.7	<0.01	RANDOM

*—uncalculated

## Data Availability

This data are subject to disclosure restrictions. Data were provided by the Slovak Breeding Services, s.e.

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
