# Peer review of "Analysis of Calving Ease and Stillbirth and Their Impact on the Length of Functional Productive Life in Slovak Holstein Cattle"

_animals, 2023, doi:10.3390/ani13091496_

Round 1

Reviewer 1 Report

This study presents a descriptive analysis of the relationship between dairy cow longevity and selected factors. In addition, survival analysis was also carried out. Cow longevity is a current issue of high interest to the industry and consumers, making the topic of the paper very timely. These are my comments and suggestions for the manuscript. 

General comment in the Introduction: There are a couple of paragraphs in the Introduction that are just single sentences. In addition, it is not clear what is the relevance of the study. Please, revise this section to concatenate the ideas from single-sentence paragraphs and make it clear what is the relevance of the study.

Line 48-49: This is a very strong causal statement. It must be referenced.

Line 69: Reference for longevity heritability?

Line 101-106: What is the percentage of right censored data in each category? What is the reason for each category? In addition, it is not clear how some of these categories are right censored. For example, how (and why) was a cow with less than 1700kg of milk censored? Add more information on the right censoring of the data.

Line 107-108: Different longevity metrics are presented in the introduction, but only the length of functional productive life was considered in this study. Why?

Line 110: “and”?

Line 112-113: Why were twin calves removed?

Line 118-119: What is the percentage of missing data? How did you handle missing data in the other variables you have in the models?

Line 121: Make it clear that the study evaluated the influence of selected traits on the longevity of the dam and not the offspring.

Line 123-126: Why the interactions between calving ease and sex, as well as stillborn and sex, were not analyzed? In addition, why two separate models instead of only one?

Line 139: previous instead of last

Line 148-151: Where is the zero category described in Lines 118-119? In addition, why did you differentiate between stillborn and died within 48 hours in presenting the descriptive stats but not in the models?

Line 152-154: It is very unclear the reasons for having “died within 48 hours of birth” and “stillborn” as the levels for the variable “sex”. It is indicated in the manuscript that these two levels have unknown sex, so a more appropriate approach is to 1) remove those observations from the analysis OR 2) create a third level called “unknown”. These changes in the data must be made and the statistical analysis re-run.

Table 1: Please improve the table caption to help the reader to understand the organization of the values in it. Currently, I find the table confusing. For instance, what are the categories of “calving ease”? (It is described in the text, but the tables/figures must be self-explanatory) Why are there only “stillborn” and “died within 48hrs” for calves “without sex determination of the calf”? Replace “Overall” with “Total”

Line 189-190: It is not clear how this sentence is contributing to the discussion here

Line 210-212: How so? Could you please elaborate on that discussion point?

Table 2: It is indicated that only cows with up to 5 lactations were available in the data. Why are there 5 and 6 parity dams in this table? In addition, it is not clear how the average values in the last column were calculated. Instead, would that be another “Total” column similar to those in Table 1?

Line 215-216: Female and male calves instead of heifers and bulls.

Table 3: What is the number of stillborns and calves that died within 48hrs with known sex?

Line 231-237: What could explain that?

Line 257-259: I would suggest moving away from referring to it as model 1 and model 2. Instead, focus on what factor (calving ease or stillbirth) was in the model. That would make it easier for the reader. The same comment goes for the whole paper (abstract, M&M, and Results & Discussion)

Line 260: What does “failure rate increases” mean in this context? Please, develop more the interpretation.

Line 265-266: Develop more what you mean by “less reliable results”.

Table 5: Improve the caption of the table. What are the values you have on the table? Is it days from birth to culling/censoring? Is it days from 1st calving to culling censoring?

Line 272: The values in between parenthesis are not the hazard ratio of culling as it is implied. It must be clear that the values are the change in -2logL. The same is valid for the remainder of the paragraph.

Table 6: Include the p-values on the table.

Line 281: Length of “functional” productive life? Also, what was the p-value that allowed you to conclude that it was significant?

Line 282-283: Write down what is the 1st category

Line 284-285: The term “easy calving” was used here, but “without assistance” was used in the figures and other parts of the manuscript. You must standardize it.

Figures 1 and 2: There must be a measure of uncertainty/error associated with the RRC in the plots. In addition, I recommend not using the line plot since it conveys an idea of “ordering” between factors when in reality the order in which male and female are presented (for example) is a choice. Please, avoid referencing Model 1 or Model 2. See my comment for lines Line 257-259

Line 291-293: This is a very important piece of information. It should be presented in the M&M section.

Line 294: higher risk of “early” culling. All cows will eventually be culled in a dairy production system. The analysis described in the manuscript highlights factors associated with the animals being culled earlier than later. This needs to be incorporated into the whole manuscript.

Line 295: Length of “functional” productive life?

Line 304-306: Are those results from this study or reported by other studies? If by others, what is the relationship with the current study?

Line 281-308: What are the implications of these results?

Line 317-318: “Difficult calving and related calf mortality”. Is that concluding that calves from a difficult birth are associated with high mortality? 

Line 325-328: This conclusion point is not supported by the current study.

Author Response

Dear Reviewer 1,

Thank you for the valuable advice and comments on our manuscript.

Simple summary and Abstract were corrected based on the changes that were made in the Manuscript according to the reviewers' requests.

References were corrected according to journal guidelines.

Response to Reviewer 1 Comments

General comment in the Introduction: There are a couple of paragraphs in the Introduction that are just single sentences. In addition, it is not clear what is the relevance of the study. Please, revise this section to concatenate the ideas from single-sentence paragraphs and make it clear what is the relevance of the study.

Response: The Introduction has been revised according to reviewers' comments.

Line 48-49: This is a very strong causal statement. It must be referenced.

Response: Line 47 – 48: „...This selection method caused unfavorable genetic responses for traits like“...was inserted instead of „....This selection method has significantly reduced...“, and this statement was referenced.

Line 69: Reference for longevity heritability?

Response: Line 61: The reference was added.

Line 101-106: What is the percentage of right censored data in each category? What is the reason for each category? In addition, it is not clear how some of these categories are right censored. For example, how (and why) was a cow with less than 1700kg of milk censored? Add more information on the right censoring of the data.

Response: In the Survival analysis, the censoring is computed for all data, not according to individual categories. The main reason for censoring is that these animals are alive at the time of analysis.

Milk yield of less than 1700 kg may bias the estimate of the effect of milk production on cow culling, therefore the data are marked as censored.

Line 95-100: Information on the right censoring was added in the MaM.

Line 107-108: Different longevity metrics are presented in the introduction, but only the length of functional productive life was considered in this study. Why?

Response: Longevity was expressed as the length of functional productive life (LFPL) because it better describes real longevity, which is not affected by voluntary culling due to low milk production.

Line 100-102: Inserted into text.

Line 110: “and”?

Response: Line 105: corrected - „and“ instead of „a“

Line 112-113: Why were twin calves removed?

Response: Based on the methodology for genetic evaluation of calving ease in Slovakia, twin calves were omitted from the data due to reduce bias.  

Line 108-109: Explanation was added in the MaM

Line 118-119: What is the percentage of missing data? How did you handle missing data in the other variables you have in the models?

Response: 37% missing data (information was added in the MaM Line 120.

For the other factors, there are incomplete data only for milk production, which is treated with censored data.

Line 121: Make it clear that the study evaluated the influence of selected traits on the longevity of the dam and not the offspring.

Response: Line 122: „The influence of selected traits on the longevity of cows was evaluated“....

Line 123-126: Why the interactions between calving ease and sex, as well as stillborn and sex, were not analyzed? In addition, why two separate models instead of only one?

Response: We evaluated each factor separately because we do not have records of the sex of the calves in stillbirth.

MaM was corrected:  Line 123-127.

We performed 2 separate analyzes for calving ease and stillbirth traits. Based on this correction, the results and the discussion are also elaborated.

Line 139: previous instead of last

Response: Line 140 – corrected - previous instead of last

Line 148-151: Where is the zero category described in Lines 118-119? In addition, why did you differentiate between stillborn and died within 48 hours in presenting the descriptive stats but not in the models?

Response: In the original manuscript: The zero category was used in the model, but in the Figure 2 and 3 was not presented. Because these groups were the reference base, the new reference levels had to be recalculated.

In the corrected manuscript: For a more comprehensible presentation of the results, we included Zero category in the Figures 2 and 3.

Missing data (zero category) were also added to MaM (Line 106-119), to the description of the model and to the results and discussion.

Stillborn and died within 48 hours were presented also in the model. (Line 155-156).

Line 152-154: It is very unclear the reasons for having “died within 48 hours of birth” and “stillborn” as the levels for the variable “sex”. It is indicated in the manuscript that these two levels have unknown sex, so a more appropriate approach is to 1) remove those observations from the analysis OR 2) create a third level called “unknown”. These changes in the data must be made and the statistical analysis re-run.

Response: In the genetic evaluation of calving ease and stillbirth in Slovakia, a methodology is used in which gender is classified in this way in order to determine whether stillbirth is the result of health or genetic problems of the cow or insufficient care of the calf after birth.

In the MaM and in the Results, we modified the text Line 119 and Table 1, 3, Figure 2,3 so that they were clearly interpreted.

Table 1: Please improve the table caption to help the reader to understand the organization of the values in it. Currently, I find the table confusing. For instance, what are the categories of “calving ease”? (It is described in the text, but the tables/figures must be self-explanatory) Why are there only “stillborn” and “died within 48hrs” for calves “without sex determination of the calf”? Replace “Overall” with “Total”

Response Table 1: Table was improved.

Line 189-190: It is not clear how this sentence is contributing to the discussion here

Response: The sentence was removed.

Line 210-212: How so? Could you please elaborate on that discussion point?

Response: Line 214-216 corrected „Many authors [26,36,39] confirm the influence of age at first calving on the incidence of difficult calvings. It's probably related to the insufficient development of the reproductive tract, and the ratio between the calf size and the feto-pelvic incompatibility [26].

Table 2: It is indicated that only cows with up to 5 lactations were available in the data. Why are there 5 and 6 parity dams in this table? In addition, it is not clear how the average values in the last column were calculated. Instead, would that be another “Total” column similar to those in Table 1?

Response: Table 2:

In the MaM, the number of lactations was incorrectly stated. The correct one is 6. In the longevity analysis, this 6th lactation is censored.

Line 91, 6 instead of 5 parities

The average values in the last column were calculated as the sum of individual parities.

Overall average was corrected to “Total”.

Line 215-216: Female and male calves instead of heifers and bulls.

Response: Line 218-219, corrected – „The frequency of born and alive female calves was 1.34% higher than that of male calves“ Instead of The frequency of born and alive heifers was 1.34% higher than that of bulls.

Table 3: What is the number of stillborns and calves that died within 48hrs with known sex?

Response Table 3: This data is not recorded during data collection. This explanation is added to MaM, Line 115-116.

Line 231-237: What could explain that?

Response: Line 236-243: This paragraph was modified.

 Line 257-259: I would suggest moving away from referring to it as model 1 and model 2. Instead, focus on what factor (calving ease or stillbirth) was in the model. That would make it easier for the reader. The same comment goes for the whole paper (abstract, M&M, and Results & Discussion)

Response: Line 262-264: This paragraph was corrected according to the reviewer's comments.

Line 260: What does “failure rate increases” mean in this context? Please, develop more the interpretation.

Response: Line 265: „the culling risk increases with time and vice versa” instead of “failure rate increases”

Line 265-266: Develop more what you mean by “less reliable results”.

Response: Line 275-276: „Many authors stated much higher censoring in Survival analysis which may lead to less reliable estimation of breeding values of longevity [9,17,41-43] due to the use of more incomplete data.“

Table 5: Improve the caption of the table. What are the values you have on the table? Is it days from birth to culling/censoring? Is it days from 1st calving to culling censoring?

Response Table 5: „Descriptive statistics for length of functional productive life“ instead of „Simply statistics for longevity“

According to the caption, it is clear that the time period was from the first calving to death or censoring.

Line 272: The values in between parenthesis are not the hazard ratio of culling as it is implied. It must be clear that the values are the change in -2logL. The same is valid for the remainder of the paragraph.

Response: Line 278-282: Paragraph was corrected in terms of the reviewer's comment.

Table 6: Include the p-values on the table.

Response Table 6: P – values were included.

Line 281: Length of “functional” productive life? Also, what was the p-value that allowed you to conclude that it was significant?

Respose: Line 288-289.

Line 282-283: Write down what is the 1st category

Response: Line 289-290: „The risk of early culling in the cows with moderately difficult calving was  1.259 times higher than in ones with easy calving (Figure 2). “  instead of origin text

Line 284-285: The term “easy calving” was used here, but “without assistance” was used in the figures and other parts of the manuscript. You must standardize it.

Response: Classes of calving ease were standardized throughout the manuscript.

Figures 1 and 2: There must be a measure of uncertainty/error associated with the RRC in the plots. In addition, I recommend not using the line plot since it conveys an idea of “ordering” between factors when in reality the order in which male and female are presented (for example) is a choice. Please, avoid referencing Model 1 or Model 2. See my comment for lines Line 257-259

Response: Figures 2 and 3 were corrected.

Line 291-293: This is a very important piece of information. It should be presented in the M&M section.

Response: Information were included in the MaM, Line 115-116.

Line 294: higher risk of “early” culling. All cows will eventually be culled in a dairy production system. The analysis described in the manuscript highlights factors associated with the animals being culled earlier than later. This needs to be incorporated into the whole manuscript.

Response: „early“ was incorporated into the whole manuscript.

Line 295: Length of “functional” productive life?

Response: Line 303, „functional“ was added

Line 304-306: Are those results from this study or reported by other studies? If by others, what is the relationship with the current study?

Response: Line 322: added text „The authors stated, that”….

Line 281-308: What are the implications of these results?

The implications of these results: The estimation of the impact of calving ease, stillbirth and sex of calf factors on longevity through the risk of early culling of cows. These factors are suitable as an earlier indirect prediction of longevity.

Line 317-318: “Difficult calving and related calf mortality”. Is that concluding that calves from a difficult birth are associated with high mortality? 

Response: Conclusion was rewrited according to the reviewer's comments. L 329-330.

Line 325-328: This conclusion point is not supported by the current study.

Response: Conclusion was rewrited according to the reviewer's comments. L 329-335.

Reviewer 2 Report

The paper is of interest in that it provides data on the relationship between dairy cow ease of calving and longevity. However as cited by the authors, it is largely confirmatory of several / many other studies in other countries.  Neverthe less it is new data for the Slovak Republic in these particular cows.

I think the English language and formatting especially in the M&M section need some work.  A few other comments:

I would have preferred it to be structured as separate Results and Discussion sections.  The references to other studies interspersed with the current data make it quite difficult to focus on the present results.  I have suggested major revision rather than 'minor' to refelect this. 

l. 263.  specialist terminology like 'right censored' might need a bit more explanation for some readers.

Author Response

Dear Reviewer 2,

Thank you for the valuable advice and comments on our manuscript.

Simple summary and Abstract were corrected based on the changes that were made in the Manuscript according to the reviewers' requests.

References were corrected according to journal guidelines.

Response to Reviewer 2 Comments

The paper is of interest in that it provides data on the relationship between dairy cow ease of calving and longevity. However as cited by the authors, it is largely confirmatory of several / many other studies in other countries.  Neverthe less it is new data for the Slovak Republic in these particular cows.

I think the English language and formatting especially in the M&M section need some work.  A few other comments:

Response: The English language was corrected. The corrected manuscript will be attached.

I would have preferred it to be structured as separate Results and Discussion sections.  The references to other studies interspersed with the current data make it quite difficult to focus on the present results.  I have suggested major revision rather than 'minor' to refelect this. 

Response: The results were elaborated and logically followed by a discussion. These chapters were keeping in their original form.

  1. 263.  specialist terminology like 'right censored' might need a bit more explanation for some readers.

Response: Terminology was explained in MaM, Line 95-100.

Reviewer 3 Report

Dear authors, your manuscript dealt with the Analysis of calving ease and stillbirth and their impact on the length of functional productive life in Slovak Holstein cattle, interesting topic, very actual. I have some comments in the attached file. best

Author Response

Dear Reviewer 3,
Thank you for your valuable advice and comments on our manuscript.

Simple summary and Abstract were corrected based on the changes that were made in the Manuscript according to the reviewers' requests.

References were corrected according to journal guidelines.

Response to Reviewer 3 Comments

Introduction:

Response: Introduction was rewritten, BCS was included into text. Line 69.

References was corrected according to journal guidelines.

MaM:

Response: Censored data contain incomplete information about the length of productive life, but they nevertheless contribute to the estimation of breeding values ​​of longevity.

These data were incorporated into analysis.

Line 152: R2

Response: corrected, Line 160.

Table 1 need editing

Response: Table 1 was edited.

Response: Figures 1 and 2 was corrected.

The conclusion was corrected.

Line 312

Response: We do not understand this comment.

Round 2

Reviewer 1 Report

The paper was improved after the first round of revision. However, I still have some comments that need to be addressed.

Abstract: Study objective is not clear in the abstract.

Line 35-38. Please revise the culling risk values. There are discrepancies between what is written and shown in the graphs.

The introduction still does not make it clear what gap this research is filling. I would suggest a sentence or two were included in the introduction stating that right before the objective is presented.

Parity 5 and 6 are combined in some places, but they are presented separately in others. This should be standardized.

Mortality if presented as “stillborn” and “died within 48 hours of birth”, “Stillborn and died after calving”, and “stillbirth”. This should be standardized as well.

Figure 1 is presented twice.

Line 290-298. Some of the risk of culling values indicated in the text do not match with those presented in the figures. This needs to be corrected.

Line 294. “… times higher for this and that, respectively, compared to easy calving.

Line 337-338. The study results do not support this first sentence.

Author Response

Dear Reviewer,

Your recommendations have been taken into account in the Manuscript. Due to the change in the order of the authors in the Introduction, the order in the References was also adjusted.

Abstract: Study objective is not clear in the abstract.

Response 1: Line: 22-24 added sentence about the aim of the study.

Line 35-38. Please revise the culling risk values. There are discrepancies between what is written and shown in the graphs.

Response 2: The values ​​1.259; 1.711 and 1.894 are correct. They mean how many times the risk of early culling a cow is higher compared to easy calving. These values ​​are calculated according to the Survival Kit analysis methodology and they are used to compare risk between individual classes. The Figures show the value of the risk of early culling for a specific class.

Line 36: added „moderately“

The introduction still does not make it clear what gap this research is filling. I would suggest a sentence or two were included in the introduction stating that right before the objective is presented.

Response 3: The paragraph: “The calving is affected by the size of the calf, parity and additional factors such as gestation length, age at first calving, and calving interval [26]. Several authors confirmed the relationship between difficult calving and calf size, and cow parity. In primiparous cows larger calves showed higher mortality rate than in cows of higher parity. This is primarily related to the size of the birth canals, which are less restrictive in older cows. It is easier to give birth to larger, better developed calves which would be more difficult in primiparous cows [17,27,28].” was moved from Line 77-83 to Line 74-81.

The paragraph: „Difficult calvings reduced the length of a productive life by 10% [21], and the risk of cow culling in difficult births increases in the first 30 days after calving [19,24]. T.he course of calving is very often an important predisposing factor for later appearing fertility disorders [25]. „ was moved from Line 73-76 to Line 82-85.

For this reason, the order of authors in The References was also corrected.

„Parity 5 and 6 are combined in some places, but they are presented separately in others. This should be standardized.

Response 4: Parity 5 and 6 were added into Table 2 separately.

Mortality if presented as “stillborn” and “died within 48 hours of birth”, “Stillborn and died after calving”, and “stillbirth”. This should be standardized as well.

Response 5: Table 3.  Stillbirth instead of Stillborn

Table 4: „Frequency of calf mortality by parity of dam. „ instead of „Frequency of stillbirth by parity of dam.“

Stillbirth instead of stillborn.

Figure 1 is presented twice.

Response 6: Figure 1 in text is presented only once. I don't see him there twice.

Line 290-298. Some of the risk of culling values indicated in the text do not match with those presented in the figures. This needs to be corrected.

Response 7: see Response 2.

Line 294. “… times higher for this and that, respectively, compared to easy calving.

Response 8: The sentence was corrected. Line 294-295.

Line 337-338. The study results do not support this first sentence.

Response 9: The sentence was corrected. Line 337-338.

Reviewer 2 Report

I am disappointed that the authors have not taken my advice about separating the Results and Discussion because I think it would have added a lot of clarity to the presentation. 

Author Response

Dear Reviewer,

We appreciate your recommendations,
but we used the option of combining Results and Discussion.
Thank you for your opinion.

Reviewer 3 Report

the paper improved a lot, i endorse the pubblication

Author Response

Thank you for your support.
